# Open-Source Artificial Intelligence System Supports Diagnosis of Mendelian Diseases in Acutely Ill Infants

**DOI:** 10.3390/children10060991

**Published:** 2023-06-01

**Authors:** Joseph Reiley, Pablo Botas, Christine E. Miller, Jian Zhao, Sabrina Malone Jenkins, Hunter Best, Peter H. Grubb, Rong Mao, Julián Isla, Luca Brunelli

**Affiliations:** 1Division of Neonatology, Department of Pediatrics, University of Utah School of Medicine, Salt Lake City, UT 84108, USA; 2Foundation Twenty-Nine, 28223 Madrid, Spain; 3Nostos Genomics, 10625 Berlin, Germany; 4ARUP Laboratories, University of Utah Health Sciences Center, Salt Lake City, UT 84108, USA; 5Valley Children’s Healthcare, Madera, CA 93636, USA; 6Department of Pathology, University of Utah School of Medicine, Salt Lake City, UT 84132, USA

**Keywords:** artificial intelligence, natural language processing, genomics, differential diagnosis, computer assisted diagnosis, electronic medical record, pediatrics, neonatal, intensive care unit

## Abstract

Mendelian disorders are prevalent in neonatal and pediatric intensive care units and are a leading cause of morbidity and mortality in these settings. Current diagnostic pipelines that integrate phenotypic and genotypic data are expert-dependent and time-intensive. Artificial intelligence (AI) tools may help address these challenges. Dx29 is an open-source AI tool designed for use by clinicians. It analyzes the patient’s phenotype and genotype to generate a ranked differential diagnosis. We used Dx29 to retrospectively analyze 25 acutely ill infants who had been diagnosed with a Mendelian disorder, using a targeted panel of ~5000 genes. For each case, a trio (proband and both parents) file containing gene variant information was analyzed, alongside patient phenotype, which was provided to Dx29 by three approaches: (1) AI extraction from medical records, (2) AI extraction with manual review/editing, and (3) manual entry. We then identified the rank of the correct diagnosis in Dx29’s differential diagnosis. With these three approaches, Dx29 ranked the correct diagnosis in the top 10 in 92–96% of cases. These results suggest that non-expert use of Dx29’s automated phenotyping and subsequent data analysis may compare favorably to standard workflows utilized by bioinformatics experts to analyze genomic data and diagnose Mendelian diseases.

## 1. Introduction

Mendelian disorders, genetic diseases attributable to a single gene, are prevalent in neonatal and pediatric intensive care unit (NICU/PICU) settings and are associated with significant morbidity and mortality [1,2,3,4]. Rapid diagnosis of Mendelian disorders can shape clinical decision-making and lead to improved outcomes [4,5,6,7]. However, these diseases often have multiple underlying etiologies and variable presentation, making them challenging to diagnose. Advances in genome-wide sequencing such as exome and genome sequencing (ES and GS) have allowed clinicians to address this challenge for patients with suspected Mendelian disorders [5,8,9]. Nevertheless, several barriers to the universal adoption of these technologies remain, including the education of frontline medical professionals, high costs, and complex data analysis processes [10,11].

ES and GS occur in two general steps: first, extraction and sequencing of DNA from the patient sample (blood, saliva, etc.); second, analysis of the sequencing data in which variants are identified by comparing the sequence to a genomic standard. The variants are then filtered according to the analysts’ parameters (e.g., population frequency) to identify those most likely to cause disease. In addition to genomic analysis, manual review of a patient’s medical records identifies key phenotypic features to guide the diagnostic process. Standard diagnostic pipelines that integrate a patient’s phenotypic and genotypic information are time-intensive; even rapid optimized pipelines can require days to weeks to result in a diagnosis [7,12,13,14]. These processes are also dependent on extensive genetics and bioinformatics expertise (Figure 1) [15]. For NICUs and PICUs not connected to academic medical centers, access to this expertise can be limited. In the United States, more than 23,000 NICU beds are spread across almost 1400 hospitals; of these hospitals, less than one-third are estimated to be affiliated with an academic institution [16,17]. Integration of artificial intelligence (AI) into these analytical pipelines is an active area of research that may help address this expertise gap [18,19,20,21].

AI has been applied to clinical genomics in a variety of ways [19,22]. One prominent example is the use of Natural Language Processing (NLP)—an AI technology that allows for analysis of unstructured text—to aid in diagnosis of patients with genetic diseases by automatically extracting phenotypic information on a patient via analysis of his/her medical records; in combination with a patient’s genetic data, automated phenotyping may reduce the amount of time between DNA sequencing and diagnosis [18,20,21]. However, many AI programs developed to date are proprietary and largely designed for use by bioinformaticists and geneticists rather than the clinicians overseeing patient care [18,20].

Dx29 (Foundation Twenty-Nine, Madrid, Spain), by contrast, is an open-source web-based AI application developed to simplify the genetic diagnostic process by making it accessible to clinicians directly caring for patients with suspected Mendelian disorders [23]. By using a patient’s phenotype (supplied manually by the clinician or extracted automatically from patient records using NLP), either alone or in conjunction with a patient’s genetic data, Dx29 generates a differential diagnosis consisting of the diseases that would best account for the patient’s clinical and molecular features. Dx29 differs from other AI applications in that it is open-source and designed for use by general clinicians rather than bioinformaticists. Moreover, Dx29 allows patients and families to collaborate with providers in the diagnostic process by uploading documents and reporting symptoms.

Accessibility of the genetic diagnostic process in NICUs and PICUs could proffer a number of benefits, including lower costs and increased utilization of new technologies such as ES and GS. It could also facilitate reanalysis of a patient as his/her clinical presentation evolves. Therefore, validation of accessible, open-source tools such as Dx29 is essential. In this study, we tested the ability of Dx29 to retrospectively identify the correct Mendelian disease in previously diagnosed acutely ill infants in the NICU and PICU. These infants were diagnosed through a typical genomic diagnostic pipeline (Figure 1), using a targeted sequencing panel called RapSeq (ARUP Laboratories, Salt Lake City, UT, USA) [7,8].

## 2. Materials and Methods

### 2.1. Participants

We retrospectively analyzed 25 trios (infant + parents) who met the following two criteria: (1) the infant was admitted to the NICU or PICU; and (2) the infant had been diagnosed with a Mendelian disorder using RapSeq, a targeted ~5000-gene panel that covers most known disease-causing genes [7,8].

### 2.2. Dx29: Technology and Data Protection

Dx29 uses NLP powered by Microsoft Text Analytics for Health (Microsoft Corporation, Redmond, WA, USA) to identify symptoms and signs in patient medical records and translate those data into Human Phenotype Ontology (HPO) terms, a hierarchy of standardized phenotypic descriptors [24,25]. To generate a differential diagnosis using both phenotypic and genotypic information, Dx29 uses Exomiser, an open-source tool [26,27]. To generate a differential diagnosis with phenotypic information alone, Dx29 compares patient phenotype to information in the Orphanet and OMIM databases to identify likely disease candidates [28,29].

Dx29 is hosted on Microsoft Azure servers, which are equipped with identity management services, threat protection, compliance tools, data privacy tools, and encryption mechanisms for stored and in-transit data to ensure the privacy and security of patient medical information. Patient data may also be withdrawn from Dx29 by the patient at any point.

### 2.3. File Types Relevant to This Study


Variant Call Format (VCF) files
○Following DNA sequencing, the raw sequencing data are compared to a genomic standard to identify the proband’s genetic variants. These variants are compiled into a VCF file. We used trio VCFs (merged data from the proband and two parents) for our genetic analysis in Dx29.Pedigree (PED) files○A PED file is a structured text document that explains the relation between multiple genetic samples. For each case, we prepared a PED file that listed the sex of each genetic sample and indicated which sample corresponded to the proband. Dx29 requires a PED file when performing trio analyses.PDF files○Dx29 can analyze and extract a patient’s phenotype from PDFs, text documents, or images of documents. All records uploaded to Dx29 for this study were PDFs, including scanned images saved as PDFs.


### 2.4. Dx29 Workflow

Dx29 is a web-based application (http://dx29.ai, accessed on 2 August 2021). After signing into the application, these are the general steps to performing an analysis on a patient’s data:Enter the case ID (required) and demographic information (optional).Provide phenotype information by performing the following:
Entering patient’s phenotype manually as HPO terms.Uploading patient medical records for automated extraction of phenotypic information.Typing a medical description of the patient and then using the automated extraction.Phenotype extraction: If medical records were uploaded as PDFs, text documents, or images of documents, Dx29 will then review each record and identify symptoms and translate them to HPO terms. This step is multilingual, supporting 50+ languages. The accuracy of the HPO identification depends on the languages, and user validation is important.A user may optionally review the extracted HPO terms within Dx29. Under each term, Dx29 will show, in context, where it was identified in the files provided. Terms deemed inaccurate or irrelevant can be removed from the subsequent analysis.Genotype analysis: The trio’s merged VCF file and corresponding pedigree file are uploaded; Dx29 then filters and annotates the variants according to preset parameters. The variants are then ranked by likelihood of causing disease based on the predicted variant pathogenicity and the clinical significance of the affected gene. Those most likely to be disease-causing are prioritized in the ranking of the final differential diagnosis.Dx29 allows some exploration of the salient variants, including type of mutation, ClinVar status, in silico pathogenicity scores, and references to relevant literature. At the time of this study, there was no functionality in Dx29 that allowed for the removal of a particular variant from consideration when building the patient’s differential diagnosis.Generation of differential diagnosis: Manually provided or automatically extracted phenotypic information is compared to the candidate variants to generate a differential diagnosis.The generated differential diagnosis consists of up to 100 diseases that are ranked by how plausibly that diagnosis could explain the patient’s symptoms and findings, while also considering how the patient’s genotype does or does not support that potential diagnosis.For each diagnosis on the list, Dx29 will show how the patient’s symptoms overlapped with the expected phenotype for that disease and show the potentially causative variant.Because HPO terms are organized hierarchically, when making comparisons between a patient’s phenotype and the expected phenotype of a disease, Dx29 can extrapolate on imperfect matches. For example, if a patient is assigned the HPO term “abnormal aortic valve morphology” (HP:0001646), which is found under “abnormal heart valve morphology” (HP:0001654) in HPO hierarchy, this is considered by Dx29 to be a match between terms.

Of note, not every disease in the ranked differential diagnosis generated in Step 5 is associated with a variant. Some diagnoses are ranked only on the basis of phenotypic overlap.

### 2.5. Study Design

Each patient’s phenotypic information (HPO terms) was uploaded to Dx29 and analyzed alone and in combination with genotypic information (VCF files). Dx29 used both the phenotypic data alone and the phenotypic and genotypic data together to generate a ranked differential diagnosis consisting of the most probable diagnoses based on the data provided.

#### 2.5.1. Phenotype Analysis

Each patient’s set of HPO terms was determined using three distinct approaches:The patient’s medical records were uploaded to Dx29, and HPO terms were extracted automatically from the text and not manually reviewed.The patient’s records were uploaded to Dx29 for automatic extraction of HPO terms, and then each term was reviewed to determine if it had been extracted in error (Section 2.4, step 3a). Terms that were incorrectly identified (e.g., “cerebral palsy” being identified in a document that uses the abbreviation “cp” for some other purpose) were then removed from the analysis. Terms were removed strictly based on whether they had been correctly identified by Dx29 from the provided records, and not on judgment of their perceived relevance to a genetic diagnosis.The HPO terms used by the RapSeq team in reaching the original diagnosis were uploaded to Dx29 without any of the patient’s medical records; these terms were generated by manual review of the patient’s records by genetic counselors in the RapSeq pipeline.

#### 2.5.2. Patient Medical Records

For the analyses that required patient medical records to be uploaded to Dx29 (Section 2.5.1, approaches 1 and 2), up to 14 days of records were used. If a patient’s stay in the NICU or PICU exceeded 14 days, records were taken from the patient’s first seven days and the seven days preceding collection of the patient’s genetic material. Only records from the patient’s time in the NICU/PICU before a genetic sample was obtained for sequencing were included. Additionally, only records of the following types were used: progress notes written by physicians or advance practice clinicians (APCs), imaging reports, history and physical documents, admission/discharge/transfer summaries, consultation notes written by physicians and APCs, and procedural findings. Notes were written by NICU/PICU providers and inpatient consulting services (e.g., genetics, surgery, and neurology). Records were downloaded as PDFs, and all potentially identifying patient information was redacted in Adobe Acrobat before uploading to Dx29. Because Dx29 is not presently equipped to process larger medical records, all files were split into fragments of about 500 KB or smaller before upload.

#### 2.5.3. Preparing for Genotype Analysis

At the time we performed this study, Dx29 required VCFs from the patient and their parents to be combined into a single VCF file and for a pedigree file to be provided alongside the VCF. We used bcftools (version 1.12-57-g0c2765b) to process and combine the VCF files from each patient and their parents and PLINK (version 1.90b6.24) to prepare a pedigree file for each trio [30,31].

### 2.6. Outcomes

Our two primary outcomes in this study were the following:The processing time for each case, beginning with creation of a patient case in Dx29, progressing through phenotypic and genotypic analysis, and ending when the ranked differential diagnosis was generated (Section 2.4, steps 1–5);How often the patient’s correct diagnosis was identified in the top 5 or top 10 potential diagnoses generated by Dx29.

## 3. Results

In a previous report, patients were recruited in the RapSeq study to assess the diagnostic capability in the NICU of a rapid genome-wide panel encompassing about 5000 genes [7,8]. The diagnostic pipeline for RapSeq is shown in Figure 1. The process is expertise-dependent and involves manual review of patient records to identify salient phenotypic features as HPO terms. HPO terms are then correlated with the variant analysis pipeline to reach a final diagnosis. Twenty-five trios previously sequenced and analyzed in the RapSeq study were selected for analysis by Dx29. The affected gene and OMIM diagnosis for each case are shown in Table 1. Of note, at the time of this study, Dx29 was not set up to predict a negative result; whether a likely candidate disease was identified or not, a differential diagnosis was generated in each case. Additionally, Dx29’s interface did not facilitate sufficient exploration of each variant and potential disease to allow us to rule out a diagnosis with this interface alone. Accordingly, only positive RapSeq cases that resulted in the identification of a variant that explained the patient’s phenotype were included.

### 3.1. Analysis with Automated Extraction of HPO Terms

In the first phase of the study, data in the form of a combined VCF and pedigree file for each trio were uploaded to Dx29, alongside selected documentation from the patient’s NICU stay, and Dx29 performed an automated analysis of the patient’s clinical records and genetic data. Once Dx29 populated the list of potential diagnoses for each patient, we determined (1) the processing time for each case and (2) how often the correct diagnosis appeared in the top 5 or 10 potential diagnoses put forward by Dx29.

The average processing time for the cases was 0.32 ± 0.21 h (mean ± SD), with the majority of the time spent on analyzing the medical records to extract HPO terms. Dx29 ranked the correct diagnosis in the top 5 diseases in the differential diagnosis in 88% (22/25) of these positive cases and in the top 10 diseases in 92% (23/25) of cases. The median, mean, minimum, and maximum ranks for the 25 cases are reported in Table 2, and the rank assigned to each individual case is shown in Table 1.

### 3.2. Analysis with Automated Extraction of HPO Terms Followed by Manual Review

For the manual review, we inspected each identified HPO term in context and removed it if it was erroneously pulled from the medical record. On average, the manual review removed 115.5 ± 27.4 (mean ± SD), or 59% ± 9% (mean ± SD), of the automatically extracted HPO terms, leaving an average of 82.8 ± 26.9 (mean ± SD) total HPO terms per case after manual review. After manual review, the patient’s correct diagnosis was in Dx29’s top 5 potential diseases in 88% (22/25) of cases and in the top 10 in 96% (24/25) of cases. The manual review process added an average of 1.66 ± 0.81 h (mean ± SD) to the processing time, for an average time of 1.96 ± 0.85 h (mean ± SD).

### 3.3. Analysis Using HPO Terms Utilized in the Standard Diagnostic Pipeline

These terms were generated by manual review of patient records at the time of the RapSeq analysis that led to the patient’s diagnosis. These terms were entered into Dx29, along with the corresponding trio VCF. With this approach, each case was assigned an average of 12.8 ± 6.3 (mean ± SD) HPO terms. Dx29 identified the correct diagnosis in the top 5 potential diagnoses in 88% (22/25) and in the top 10 in 92% (23/25).

### 3.4. Analysis Using Only the Patient’s Phenotype

For each positive case, we set up Dx29 to automatically extract the patient’s HPO terms from their medical records but did not upload the accompanying VCF file. The correct diagnosis appeared in the list of possible diseases generated by Dx29 in 52% (13/25) of cases (Dx29 lists up to 100 potential diagnoses). The correct diagnosis appeared among the top 5 diseases in 12% (3/25) of cases and in the top 10 diseases in 24% (6/25) of cases.

## 4. Discussion

Standard diagnostic pipelines for the analysis of genome-wide sequencing data and diagnosis of rare genetic diseases remain complex and often dependent on extensive genetics and bioinformatics expertise [15]. Dx29 is an AI-based clinical tool developed to streamline this process by rapidly identifying a list of genetic diseases potentially associated with the clinical presentation of a patient. Our data suggest that the fully automated deployment of Dx29 or similar tools can efficiently and accurately identify a 10-disease short list of potential diagnoses in cases with a positive genetic diagnosis. Further evolution of these systems could provide a new paradigm to support clinicians in the NICU and PICU who care for patients potentially affected by a genetic disease.

Standard diagnostic pipelines are not only complex but also time-consuming, with turnaround times ranging, at best, from days to weeks [7,12,13,14,20]. The clinical and analytical teams involved in the original RapSeq investigation did not track time from the moment in which data analysis started to when the diagnosis was identified; however, we estimate at least 6–12 h to be typically needed for this process. As the fully automated analysis with Dx29 required an average of <1 h, including data upload, phenotype and genotype analysis, and generation of the final differential diagnosis, this approach might help clinicians to quickly identify a manageable list of potential diagnoses among the about 7000 rare genetic diseases reported so far.

A manual reanalysis of HPO terms following automated extraction by Dx29 prolonged the time necessary for the generation of the ranked differential diagnosis. However, the time remained <3 h on average. Although our data suggest that manual reanalysis may increase the accuracy of Dx29, the improvement was limited. There was no difference in the identification of the correct diagnosis in the top 5 diseases, with 88% in both circumstances, and while we observed a marginal improvement in the top 10 diseases, from 92% to 96% of cases, this will need to be confirmed in future studies with larger cohorts. Therefore, the data we present here support the automated utilization of Dx29 and confirm the effectiveness of the algorithms it uses.

We report here that the manual review of automatically extracted phenotypes resulted in the removal of 59%, on average, of the HPO terms extracted by Dx29. As was noted, this afforded only a modest improvement in the ranking of the correct diagnosis in the final list of candidate diseases. While many of the terms that we removed were mistakes by the NLP algorithms employed by Dx29 (e.g., “cerebral palsy” being identified in a document that uses the abbreviation “cp” for some other purpose), other erroneous terms were identified out of context; for example, commentary on a patient’s chin in the physical exam resulted in Dx29 extracting the HPO term “abnormality of the chin.” That Dx29 was able to perform nearly as well when these erroneous terms were included in the analysis as when they were removed from consideration highlights how its algorithms focus on the phenotypic features most important for diagnosis of a given genetic disease.

This is further illustrated by the fact that many terms that were correctly pulled from patient records were not necessarily related to the patient’s eventual diagnosis. For example, a patient experiencing respiratory distress will be correctly assigned the corresponding HPO term by Dx29, even when this may be more relevant to the patient’s preterm birth than his/her genetic syndrome. However, we saw no improvement in our outcomes when the HPO terms that were identified by manual review of patient records and subsequently used in the standard diagnostic pipeline were provided to Dx29 in place of the automatically extracted terms.

When we used Dx29 to generate a differential diagnosis based solely on automated patient phenotype extraction from medical records, only 13 of the 25 cases had the correct diagnosis listed at any rank in the differential, which contained up to 100 diseases, and 6 cases had the correct diagnosis in the top 10. While Dx29 allows for a phenotype-only approach, these data, contrasted with the analyses that included VCF files, demonstrate the impact of using genetic and phenotypic information in tandem when analyzing complex patients.

This work has several limitations. A large limitation of Dx29’s functionality is its inability to return a “negative” result. Whether or not there are strong candidate diseases to explain the patient’s phenotype and genotype, a ranked differential diagnosis of up to 100 candidate diseases is generated, without clear criteria or metrics for excluding specific diagnoses. Given that approximately 50 percent of RapSeq cases are “negative,” using Dx29 may lead clinicians who are inexperienced in genetics into a time-consuming and possibly costly exclusion process for multiple potential diagnoses. Thus, this study demonstrates ascertainment bias because we included only patients with a “positive” result in the RapSeq study. Additionally, the ability to scrutinize variants identified in the patient’s VCF is limited, which can further limit patient analysis using Dx29 alone; for instance, without knowing the quality metrics of identified variants, it would be difficult to determine if a specific variant was likely causative. Dx29 is not designed or currently equipped to make or suggest a final diagnosis on its own; because of this and the abovementioned intrinsic limitations, performing a prospective analysis with Dx29 would still require pairing it with a conventional bioinformatics pathway to reach a diagnosis. However, the ability of Dx29 to highly rank complex patients’ correct diagnoses is promising and suggests that further development of AI tools such as Dx29 may make genomic analysis by clinicians a future reality.

In using a targeted gene panel as the source of the genetic data for this study, which resulted in VCF files of only a few megabytes, we were not able to assess how quickly Dx29 functions when larger genetic files are uploaded; additionally, in its current iteration, Dx29 is not able to analyze VCFs that are larger than about 1 gigabyte, which could limit its utility in the analysis of WES and WGS data. Furthermore, the software utilized in Dx29 to process the genomic data (Exomiser) is not currently equipped to analyze copy number variants or other large chromosomal abnormalities [27].

This study focused on infants in critical care settings (NICU and PICU) and excluded older patients and those sequenced in ambulatory or inpatient but non-critical care settings. Future research is needed to determine how Dx29 or similar tools would benefit genetic diagnosis for these patients. Similarly, Dx29 supports direct patient or family input of symptoms into the diagnostic pipeline. Further prospective research is needed to investigate the impact of this intriguing capability on patient and provider engagement, satisfaction, and trust in the results afforded by genomic diagnosis.

Despite these limitations, our study further illustrates the potential role of AI in aiding the diagnosis of complex genetic diseases in infants [18,20,21]. Use of Dx29 by non-bioinformaticists yielded promising results, showing how Dx29 and similar AI platforms may one day support genomic analysis by primary clinicians, with less dependance on bioinformatics expertise, thereby streamlining the diagnostic process.

## Figures and Tables

**Figure 1 children-10-00991-f001:**
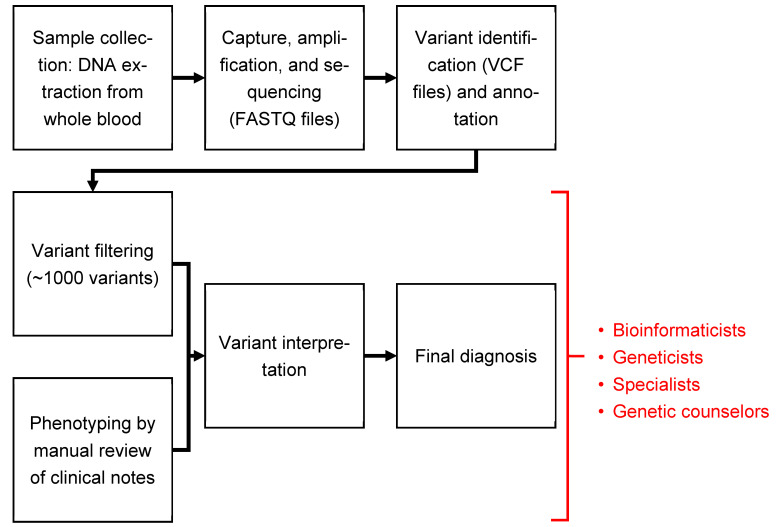
Overview of a typical genomic diagnostic pipeline, from sample collection to final results. DNA extracted from whole blood is sequenced, generating a FASTQ file. Variants, typically thousands of them, are then identified by comparing the sequencing data to a genomic standard. These variants are compiled into a VCF file and then filtered and analyzed to identify those that are most likely to cause disease. The results are narrowed down to a diagnosis by incorporating the patient’s phenotypic or symptomatic information, which is obtained through a manual review of clinical records.

**Table 1 children-10-00991-t001:** Affected gene and corresponding OMIM diagnosis for each of the 25 positive cases included in this analysis. Also included are the rank assigned to the correct diagnosis in Dx29’s differential diagnosis for each of the 4 approaches used in this study to analyze the participants: genetic analysis + automatic extraction of HPO terms from medical records, genetic analysis + automatic extraction followed by review of the extracted terms, genetic analysis + providing Dx29 with the same terms used in the original RapSeq pipeline, and using only the phenotype data from automatic extraction (no accompanying genetic analysis).

ID	Gene	Diagnosis/OMIM Diseases	Automatic	Automatic with Review	RapSeq Terms	Phenotype Only
001	CHAT	Congenital presynaptic myasthenic syndrome 6	31	31	30	20
004	FNLA	X-linked periventricular nodular heterotopia	14	6	4	unranked
005	FANCB	X-linked VACTERL with hydrocephalus syndrome	2	2	2	42
007	KMTD2	Kabuki syndrome 1	1	1	2	unranked
009	CHD7	CHARGE syndrome	1	1	1	2
013	ASXL1	Bohring–Opitz syndrome	1	1	1	8
014	FBN1	Neonatal Marfan	1	1	1	3
023	PAX3	Craniofacial–deafness–hand syndrome, Waardenburg syndrome, type 1 and type 3	1	1	1	unranked
026	CACNA1A	Developmental and epileptic encephalopathy 42	5	4	1	unranked
027	KCNQ2	Early infantile epileptic encephalopathy 7	1	1	1	44
028	HDAC8	Cornelia de Lange syndrome 5	1	1	1	1
029	AHCY	Hypermethioninemia with deficiency of S-adenosylhomocysteine hydrolase	1	1	1	60
035	ACAD9	Mitochondrial complex I deficiency, nuclear type 20	1	1	3	43
036	CDAN1	Dyserythropoietic anemia, congenital, type Ia	3	3	1	60
037	AMER1	Osteopathia striata with cranial sclerosis	1	1	2	unranked
041	TCIRG1	Osteopetrosis 1	1	1	1	9
042	RYR1	Autosomal recessive and autosomal dominant congenital neuromuscular disease with uniform type 1 fiber and with central core disease	3	1	1	unranked
044	RYR1	Autosomal recessive and autosomal dominant congenital neuromuscular disease with uniform type 1 fiber and with central core disease	5	1	1	unranked
045	GUSB	Autosomal recessive mucopolysaccharidosis VII	1	2	2	unranked
047	ASNS	Asparagine synthetase deficiency	7	6	2	57
050	CHD7	CHARGE syndrome	1	1	1	unranked
054	ACTA1	Unspecified myopathy	1	1	8	unranked
058	KCNQ3	Early infantile epileptic encephalopathy 7	1	1	1	unranked
068	ASXL1	Bohring–Opitz syndrome	1	1	2	7
069	SLC35A2	Congenital disorder of glycosylation, type IIm	1	2	11	unranked

**Table 2 children-10-00991-t002:** How Dx29 ranked the correct diagnosis in each of the three approaches: automatic extraction of HPO terms from medical record, automatic extraction followed by manual review of extracted terms, and providing Dx29 with the same terms used in the original RapSeq pipeline. Average turnaround time was defined as beginning at patient instantiation in Dx29 and ending with the generation of Dx29’s differential diagnosis.

	Automatic Extraction of Phenotype	Automatic Extraction + Manual Review	RapSeq Phenotypic Terms	Phenotype Only
Correct diagnosis is in top 5 suggested diseases	88%	88%	88%	12%
Correct diagnosis is in top 10 suggested diseases	92%	96%	92%	24%
Median rank of correct diagnosis	1	1	1	
Mean rank of correct diagnosis	3.48	2.92	3.28	
Minimum rank of correct diagnosis	1	1	1	
Maximum rank of correct diagnosis	31	31	30	
Average time elapsed (mean ± SD) in hours	0.32 ± 0.21	1.96 ± 0.85		

## Data Availability

The data presented in this study are available upon request from the corresponding author. The data are not publicly available due to privacy restrictions.

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
