# Peer review of "Open-Source Artificial Intelligence System Supports Diagnosis of Mendelian Diseases in Acutely Ill Infants"

_children, 2023, doi:10.3390/children10060991_

Round 1
Reviewer 1 Report
This manuscript presents an interesting evaluation of a free tool (Dx29) to support and fasten diagnosis of mendelian conditions in NICU/PICU, capturing phenotype data (using different approach) and genotype data (RapSeq).
The overall study design is simple but accurate. Following a couple of suggestions.
INTRODUCTION
Line 54-57: please provide a reference.
FIGURE 1:
Does the variant identification step, includes also variant annotation?
As a general concern, I was wondering to introduce, maybe in the methods section, additional information on the privacy/data protection aspects of Dx29 tool. Uploading on a website a large amount of phenotypic and genotypic data gives me some perplexities. I also propose to consider how privacy can impact on the present study, facing this topic along the manuscript.
Author Response
INTRODUCTION
Line 54-57: please provide a reference.
- Thank you for this suggestion. While the exact numbers for numbers of NICUs and their attachment to academic medical centers are difficult to come by, we offered this statistic as an estimate of how many NICUs may not have the same degree of genetic expertise as those tied to academic institutions. In response to your suggestion, we edited those lines to better represent the available data. They now say: “In the United States, more than 23,000 NICU beds are spread across almost 1,400 hospitals; of these hospitals, less than one third are estimated to be affiliated with an academic institution. Integration of artificial intelligence (AI) into these analytical pipelines is an active area of research that may help address this expertise gap.” We also added the appropriate citations.
FIGURE 1:
Does the variant identification step, includes also variant annotation?
- Thank you for this clarifying question. We have added “and annotation” to the variant identification box.
As a general concern, I was wondering to introduce, maybe in the methods section, additional information on the privacy/data protection aspects of Dx29 tool. Uploading on a website a large amount of phenotypic and genotypic data gives me some perplexities. I also propose to consider how privacy can impact on the present study, facing this topic along the manuscript.
- Thank you for bringing this to our attention. To address the security and privacy of patient data in Dx29, we added the following paragraph to the methods section (section 2.2): “Dx29 is hosted on Microsoft Azure servers, which are equipped with identity management services, threat protection, compliance tools, data privacy tools, and encryption mechanisms for stored and in-transit data to ensure the privacy and security of patient medical information. Patient data may also be withdrawn from Dx29 by the patient at any point.”
- Regarding addressing the issue of privacy throughout the manuscript, we touched briefly on how we protected patient information in the “2.5.2 Patient medical records” section of the manuscript. We redacted all potentially identifying patient information prior to data upload, so there was never concern, in the present study, for loss of patient privacy. However, as mentioned above, Dx29 is built on Azure, and is equipped with encryption and other services that protect patient data.
Reviewer 2 Report
Comments to the Authors:
The authors present a study about the use of artificial intelligence tool in reaching a genetic diagnosis in suspected Mendelian disorder in shorter duration and without the need to go through conventional path with the help of experts in the field of genetics including bioinformatics. They have assessed the reliability/sensitivity of Dx29 tool in reaching an accurate diagnosis in a retrospective 25 cases that were genetically diagnosed using conventional genetic testing methods (using targeted panel of 5,000 genes).
The article is well-written, and data is clearly presented. It adds to the knowledge and value of the use of artificial intelligence in the field of Genetics.
There are very few points to be considered which would be beneficial for this paper.
- How about the other genetic disorders that could only be diagnosed through chromosomal studies such as karyotype/SNP microarray such as chromosomal aneuploidy or copy number variants? Would Dx29 be able to give differential diagnosis for such disorders?
- Dx29 could also be used by patients/parents to diagnose their children, not restricted to clinicians as mentioned in the manuscript. Do you think that would be a problem for patient to start using such tools?
- Was the phenotype described in medical records based on the evaluation of geneticists? Or general pediatricians/ICU/PICU physicians?
- Do you recommend using Dx29 with or without genetic testing? For example, would you start with only phenotype in Dx29 and based on the differential, you would decide what genetic testing to order? For example, certain gene panels?
- Line 295: The data analysis took less than an hour. How long does it take including uploading medical record, VCF files, pedigree? This should take more than an hour.
- AI is a good tool to go with in parallel with the conventional bioinformatics method. It is useful in getting a fast potential diagnosis; however, it is not confirmatory, and you would still need to get the final conventional genetic testing results for proper diagnosis
- Line 328: Another limitation of the study that you have not tried uploading WES or WGS VCF file, rather you uploaded targeted gene panel. Have you tried including VCF for WES or WGS? Would it be able to upload such big files?
- Line 338: Classification of the pathogenicity of the variant is an important aspect in reaching a diagnosis. Would Dx29 be able to do that? Or would it include benign variants as potential candidate genes?
Author Response
There are very few points to be considered which would be beneficial for this paper.
- How about the other genetic disorders that could only be diagnosed through chromosomal studies such as karyotype/SNP microarray such as chromosomal aneuploidy or copy number variants? Would Dx29 be able to give differential diagnosis for such disorders?
- Thank you for asking this clarifying question. Exomiser, the software behind Dx29’s genomic analysis, is not able to analyze CNVs or other large chromosomal abnormalities. To address this in the manuscript, we added the following phrase to the limitations in the discussion section: “Furthermore, the software utilized in Dx29 to process the genomic data (Exomiser) is not currently equipped to analyze copy number variants or other large chromosomal abnormalities.”
- Dx29 could also be used by patients/parents to diagnose their children, not restricted to clinicians as mentioned in the manuscript. Do you think that would be a problem for patient to start using such tools?
- Thank you for bringing this up; it’s a very interesting question to consider in the age of greater access to AI (like ChatGPT). One interesting feature of Dx29 that we did not mention in the paper is the ability for providers and patients/families to collaborate through Dx29. Patients and families can upload documentation and report additional symptoms. In this way, even though this feature doesn’t preclude the ability of patients and families to use Dx29 independently from a clinician, Dx29 becomes a tool that helps to build the therapeutic alliance, rather than a resource that arms patients with extensive differential diagnoses filled with rare diseases that may end up causing more confusion than help. In our view, Dx29 would be no more likely to cause difficulty for patients and physicians in this way than many resources that are already available and can provide patients with similar results, including the OMIM webpage, which allows users to search by symptoms and then shows a list of rare diseases that match the symptoms.
- To briefly address the question, we added a sentence in the introduction about the collaborative features of Dx29: “Moreover, Dx29 allows patients and families to collaborate with providers in the diagnostic process by uploading documents and reporting symptoms.” We also briefly address your question in the discussion: “Similarly, Dx29 supports direct patient or family input of symptoms into the diagnostic pipeline. Further prospective research is needed to investigate the impact of this intriguing capability on patient and provider engagement, satisfaction, and trust in the results afforded by genomic diagnosis.”
- Was the phenotype described in medical records based on the evaluation of geneticists? Or general pediatricians/ICU/PICU physicians?
- Thank you for this clarifying question. To elucidate, we have added the following phrase to section 2.5.2 of the paper: “Notes were written by NICU/PICU providers, as well as inpatient consulting services (e.g., genetics, surgery, neurology).” This section describes which records were used for automatic extraction of patient phenotype from medical records.
- Do you recommend using Dx29 with or without genetic testing? For example, would you start with only phenotype in Dx29 and based on the differential, you would decide what genetic testing to order? For example, certain gene panels?
- Thank you for asking this. Our analysis showed a major benefit to integrating the phenotype and genotype of a patient compared to using only the phenotype. The phenotype-only approach often excluded the correct diagnosis, or ranked it low on the differential. The catch is that these patients were complex and critically ill, so the phenotype is messy. It’s possible that in simpler patients, Dx29 could be used as you describe, but this isn’t something we looked at in this study. To emphasize the difference between using phenotype alone and genotype + phenotype, we added the following paragraph to the discussion: “When we used Dx29 to generate a differential diagnosis based solely on automated patient phenotype extraction from medical records, only 13 of the 25 cases had the correct diagnosis listed at any rank in the differential, which contained up to 100 diseases, and six cases had the correct diagnosis in the top 10. While Dx29 allows for a phenotype-only approach, these data, contrasted with the analyses that included VCF files, demonstrate the impact of using genetic and phenotypic information in tandem when analyzing complex patients.”
- Line 295: The data analysis took less than an hour. How long does it take including uploading medical record, VCF files, pedigree? This should take more than an hour.
- Thank you for pointing this out. The <1 hour did in fact include the time it took to upload all patient data, perform phenotype extraction and genomic analysis, and generate the differential diagnosis. Our files were not generally very large (VCFs were a few megabytes at most), which contributed to this speedy processing time. To clarify this point in the discussion, we edited the sentence in question to say: “As the fully automated analysis with Dx29 required an average of <1 hour, including data upload, phenotype and genotype analysis, and generation of the final differential diagnosis, this approach might help clinicians to quickly identify a manageable list of potential diagnoses among the about 7,000 rare genetic diseases reported so far.”
- AI is a good tool to go with in parallel with the conventional bioinformatics method. It is useful in getting a fast potential diagnosis; however, it is not confirmatory, and you would still need to get the final conventional genetic testing results for proper diagnosis
- Yes, thank you, this is an important point to clarify. To emphasize it, we edited part of the discussion on limitations of the study to say: “Dx29 is not designed or currently equipped to make or suggest a final diagnosis on its own; because of this and the above-mentioned intrinsic limitations, performing prospective analysis with Dx29 would still require pairing it with a conventional bioinformatics pathway to reach a diagnosis”
- Line 328: Another limitation of the study that you have not tried uploading WES or WGS VCF file, rather you uploaded targeted gene panel. Have you tried including VCF for WES or WGS? Would it be able to upload such big files?
- This is an excellent point, and one which was not addressed in the manuscript. To remedy this, we added the following to the discussion on limitations of our study: “In using a targeted gene panel as the source of the genetic data for this study, which resulted in VCF files of only a few megabytes, we were not able to assess how quickly Dx29 functions when larger genetic files are uploaded; additionally, in its current iteration, Dx29 is not able to analyze VCFs that are larger than about 1 gigabyte, which could limit its utility in the analysis of WES and WGS data.”
- Line 338: Classification of the pathogenicity of the variant is an important aspect in reaching a diagnosis. Would Dx29 be able to do that? Or would it include benign variants as potential candidate genes?
- Thank you for the question. The answer, technically, would be that benign variants could end up on the differential diagnosis. However, when the variants are being analyzed, in silico pathogenicity predictors and consideration of the clinical significance of the affected gene are used to assign a prioritization score to each variant, with those more likely to be disease-causing taking priority over those more likely to be benign. When building the final differential, the predicted pathogenic variants would show up higher on the differential. Additionally, Dx29’s interface allows for a brief exploration of the variants in question once the differential diagnosis is generated. This includes ClinVar classification. So if a known benign gene somehow ended up on Dx29’s differential, the user could quickly remove the gene from consideration based on the ClinVar designation. To clarify how the variants are prioritized, we edited the methods section that describes Dx29’s workflow (section 2.4) to say the following: “Genotype analysis: The trio’s merged VCF file and corresponding pedigree file are uploaded, Dx29 filters and annotates the variants according to preset parameters. The variants are then ranked by likelihood of causing disease based on the predicted variant pathogenicity and the clinical significance of the affected gene. Those most likely to be disease-causing are prioritized in the ranking of the final differential diagnosis.”